# Efficiency of Platelet Transfusion in Patients with Moderate-to-Severe Chronic Kidney Disease and Thrombocytopenia

**DOI:** 10.3390/ijms242115895

**Published:** 2023-11-02

**Authors:** Sevigean Ali, Mihaela Botnarciuc, Lavinia Carmen Daba, Sorina Ispas, Alina Mihaela Stanigut, Camelia Pana, Marian-Catalin Burcila, Liliana-Ana Tuta

**Affiliations:** 1Preclinical Disciplines Department, Faculty of Medicine, Campus B, Ovidius University of Constanta, Aleea Universitatii nr. 1, 900470 Constanta, Romania; sevigean.ali@365.univ-ovidius.ro (S.A.); sorina.ispas@univ-ovidius.ro (S.I.); 2Blood Transfusions Unit, Emergency Clinical County Hospital Constanta, Bdul Tomis nr. 145, 900591 Constanta, Romania; 3Clinical Medical Disciplines Department, Faculty of Medicine, Campus B, Ovidius University of Constanta, Aleea Universitatii nr. 1, 900470 Constanta, Romania; alina.stanigut@365.univ-ovidius.ro (A.M.S.); camelia.pana@365.univ-ovidius.ro (C.P.); marian.burcila@365.univ-ovidius.ro (M.-C.B.); tuta.liliana@univ-ovidius.ro (L.-A.T.); 4Nephrology Department, Emergency Clinical County Hospital Constanta, Bdul Tomis nr. 145, 900591 Constanta, Romania

**Keywords:** platelets, transfusion, chronic kidney disease, uremic toxins

## Abstract

There have been relatively few studies revealing a decreased platelet count in chronic kidney disease (CKD). Although this hematological abnormality is not as well documented as renal anemia, platelet functions are altered in the uremic environment and there is an increased risk of bleeding. The aim of this study was to assess the effectiveness of the administration of platelet concentrate in CKD based on how patient prognosis was influenced by platelet transfusion therapy. The study monitored 104 patients with CKD and thrombocytopenia who received platelet transfusion during their hospitalization in the period from 2015 to 2021. The complete blood cell count, serum urea and creatinine, and inflammatory status were tested upon admission. The number of transfused platelet units were considered for each patient. A Kruskal–Wallis H test showed that for one transfused platelet unit, the distribution of the number of platelets (×10^3^/µL) was the same across the categories of associated diagnoses, which was seen as possible risk factors for thrombocytopenia, including liver cirrhosis and urosepsis. With a single exception, all patients exceeded the critical threshold of 20 × 10^3^/µL and 14 patients remained under 50 × 10^3^/µL. Even though our patients exceeded the critical threshold of platelet numbers, in patients with multiple comorbidities, severe, uncontrolled hemorrhages could not be prevented in 4.83% of cases.

## 1. Introduction

Over the past 25 years, chronic kidney disease (CKD) has become an emerging global public health problem. This chronic, non-communicable disease with increasing age-standardized death rates presents a considerable burden on global healthcare resources, necessitating costly renal replacement therapy either via dialysis or kidney transplantation. After the definition and classification of chronic kidney disease (CKD) by the Kidney Disease Outcomes Quality Initiative (KDOQI) for the first time in 2002, which was subsequently endorsed by KDIGO in 2004 and 2012, there have been increased attention to CKD in clinical practice, research, and public health [1,2]. This has generated publications that have not only debated the best estimated glomerular filtration rate (eGFR) formula and thresholds for people of different ages, races, and gender, but also the threshold of albuminuria for an accurate appreciation of the kidney damage [3,4]. The position of KDOQI and all international nephrology associations and societies have revealed that a comprehensive analysis of cardiovascular and all-cause mortality and kidney outcomes according to eGFR and albuminuria is still needed to answer key questions regarding this special population group, as researchers have discovered an association between inflammation, malnutrition, and cardiovascular disease in patients with CKD [1].

Some studies have revealed a decrease in platelet count in CKD patients, although it is not as well documented as in renal anemia [5,6]. Kidney failure is associated with blood loss, reduced red blood cell (RBC) lifespan, and impaired erythropoiesis, mainly due to erythropoietin deficiency. Another hematological abnormality discovered in CKD patients, especially in advanced stages, is a high ratio of neutrophil and lymphocyte concentration that indicates the inflammatory state of CKD. Moreover, it is known that “excessive clotting and bleeding in chronic kidney disease is related to platelet-dependent mechanisms” [7,8]. Platelet functions are “altered in the advanced phases of the disease. Platelet hyporeactivity mediated by uremic toxins and anemia contributes to the increased bleeding risk in advanced chronic kidney disease”. The activation of platelets, with the “increased formation of platelet-leukocyte conjugates, and platelet-derived microparticles”, are considered main pathogenic mechanisms of thrombosis in these patients [7,8].

This is a descriptive study of thrombocytopenic CKD patients requiring platelet transfusions. The aim of the study was to establish if the administration of the platelet concentrate was efficient and how the patients’ prognosis was influenced by platelet transfusion therapy. An important issue was to determine the platelet count threshold for prophylactic administration in patients with moderate-to-severe chronic kidney disease and thrombocytopenia, which is related to their higher risk of bleeding.

## 2. Results

### 2.1. Distribution upon Age and Gender

In the studied group, there were 55 (52.8%) men and 49 (47.12%) women. The mean age of the patients with CKD and thrombocytopenia was 63.96 years. There were statistical differences between the mean age values in males (59.75 years) and females (68.69 years) (t = −2.943, df = 98.191, *p* = 0.004 < α = 0.05).

The results of gender and age as risk factors for the progression of chronic kidney disease showed that risk was higher in men at a lower mean age (Table 1).

### 2.2. Associated Diagnoses in Patients with CKD and Thrombocytopenia: End Stage Kidney Disease Patients

Associated diagnoses, which were seen as possible risk factors for thrombocytopenia in the studied patients with chronic kidney disease (CKD), included liver cirrhosis (CKD + LC) due to hepatitis C virus (23 cases, 22.12%) and urosepsis, including recurrent urinary tract infections and kidney abscesses (23 cases, 22.12%) (CKD + S). Urinary stones or urinary obstructions were not present in our group of patients.

Among the 104 patients with chronic kidney disease and thrombocytopenia, 21 (20.19%) were end-stage kidney disease treated via hemodialysis, which is an important risk factor for thrombocytopenia (Figure 1).

There was no association between the Gender and Associated diagnosis (Χ^2^_calc_ = 0.366, df = 2, *p* = 0.833 > α = 0.05) (Table 2). Also, we found that there are no statistical differences between the mean age values according to the associated diagnosis groups (F = 1.309, df_BG_ = 2, df_WG_ = 100, *p* = 0.275 > α = 0.05) (Table 3).

### 2.3. Transfusion Requirements 

The patients’ platelets median number at admission was 49.22 × 10^3^/μL. Platelet concentrate was requested in all the studied patients.

As standard platelet concentrate units and apheresis platelet units were administrated, we equivalated one apheresis platelet unit with three standard platelet concentrated units (SPC) to have a unitary approach on transfusion. The number of transfused platelet units to the patients in the study can be synthetized as follows: 56 (53.85%) patients received one apheresis platelet unit, which was considered a low dose; 40 (38.46%) patients received two apheresis platelet units, which was considered an intermediate dose; and eight patients (7.69%) received three apheresis platelet units, which was considered a high dose (Table 4).

Even though the number of platelets available for administration was smaller than based on necessity, patients with platelet counts under the threshold of 20 × 10^3^/µL were administrated at least two apheresis platelet units per patient, considering the additional risk factors for bleeding.

### 2.4. Results of Platelet Transfusion

The results of platelet transfusion were that, with a single exception, all patients exceeded the critical threshold of 20 × 10^3^ platelets/µL. Most of the patients measured over 50 × 10^3^ platelets/µL; there were 51 cases between 50–100 × 10^3^ platelets/µL and 39 cases over 100 × 10^3^ platelets/µL. Only 14 patients remained under 50 × 10^3^ platelets/µL (Table 5, Figure 2).

Further, we analyzed the association between the associated diagnoses and number of transfused apheresis platelet units (Table 6).

We found that there was no association between the number of transfused platelet units and the presence of associated diagnoses (Χ^2^_calc_ = 7.921, df = 4, *p* = 0.095 > α = 0.05). For each associated diagnosis and number of transfused platelet units, a Wilcoxon signed-rank test elicited a statistically significant change in the number of platelets (×10^3^/μL) between the pre-transfusion and post-transfusion time points (see Table 7: z values and *p* < α = 0.05). 

Indeed, the median number of platelets (×10^3^/μ) increased from pre- to post-transfusion in each case (Table 7).

A Kruskal–Wallis H test showed that for one transfused platelet unit, the distribution of the no. of platelets (×10^3^/μL) was the same across the categories of associated diagnoses for both the pre- (H = 1.410, df = 2, *p* = 0.494 > α = 0.05) and post-transfusion (H = 4.521, df = 2, *p* = 0.104 > α = 0.05) time points. For two transfused platelet units, the distribution of the no. of platelets (×10^3^/μL) was not the same across the categories of associated diagnoses for both the pre- (H = 6.836, df = 2, *p* = 0.032 < α = 0.05, with a mean rank no. of platelet score of 24.65 for CKD + LC, 10.50 for CKD + S, and 21.30 for CKD) and post-transfusion time points (H = 6.819, df = 2, *p* = 0.033 < α = 0.05, with a mean rank no. of platelet score of 20.50 for CKD + LC, 10.57 for CKD + S, and 23.98 for CKD) (Figure 3).

A Kruskal–Wallis H test showed that, for one transfused platelet unit, the distribution of creatinine (mg/dL) was the same across the categories of associated diagnoses (H = 2.495, df = 2, *p* = 0.287 > α = 0.05) and the median values of creatinine (mg/dL) were the same across the categories of associated diagnoses (Χ^2^_calc_ = 1.851, df = 2, *p* = 0.396 > α = 0.05). This was the same for patients with two transfused platelet units: the distribution of creatinine (mg/dL) was the same across the categories of associated diagnoses (H = 0.043, df = 2, *p* = 0.979 > α = 0.05), and the median values of creatinine (mg/dL) were the same across the categories of associated diagnoses (Χ^2^_calc_ = 1.635, df = 2, *p* = 0.441 > α = 0.05) (Figure 4).

A Kruskal–Wallis H test showed that for one transfused platelet unit, the distribution of BUN (mg/dL) was the same across the categories of associated diagnoses (H = 1.047, df = 2, *p* = 0.592 > α = 0.05) and the median values of BUN (mg/dL) were the same across the categories of associated diagnoses (Χ^2^_calc_ = 4.404, df = 2, *p* = 0.111 > α = 0.05). This was the same for patients with two transfused platelet units: the distribution of BUN (mg/dL) was the same across the categories of associated diagnoses (H = 0.405, df = 2, *p* = 0.817 > α = 0.05) and the median values of BUN (mg/dL) were the same across the categories of associated diagnoses (Χ^2^_cal_ = 3.409, df =2, *p* = 0.182 > α = 0.05), as presented in Figure 5.

A Kruskal–Wallis H test showed that for one transfused platelet unit, the distribution of ALT (U/L) was the same across the categories of associated diagnoses (H = 1.925, df = 2, *p* = 0.382 > α = 0.05) and the median values of ALT (U/L) were the same across the categories of associated diagnoses (Χ^2^_calc_ = 2.071, df = 2, *p* = 0.355 > α = 0.05). The same was found for patients with two transfused platelet units: the distribution of ALT (U/L) was the same across the categories of associated diagnoses (H = 2.304, df = 2, *p* = 0.316 > α = 0.05) and the median values of ALT (U/L) were the same across the categories of associated diagnoses (Χ^2^_calc_ = 1.978, df = 2, *p* = 0.372 > α = 0.05), as presented in Figure 6.

A Kruskal–Wallis H test showed that for one transfused platelet unit, the distribution of CRP (mg/dL) was the same across the categories of associated diagnoses (H = 0.711, df = 2, *p* = 0.701 > α = 0.05) and the median values of CRP (mg/dL) were the same across the categories of associated diagnoses (Χ^2^_calc_ = 0.451, df = 2, *p* = 0.798 > α = 0.05). The same was found for patients with two transfused platelet units: the distribution of CRP (mg/dL) was the same across the categories of associated diagnoses (H = 3.555, df = 2, *p* = 0.169 > α = 0.05) and the median values of CRP (mg/dL) were the same across the categories of associated diagnoses (Χ^2^_calc_ = 4.448, df = 2, *p* = 0.108 > α = 0.05) (Figure 7).

A major change in the number of platelets (×10^3^/μL) between the pre-transfusion and post-transfusion platelet counts was noticed in the eight patients who received three apheresis platelet units (the high dose). The median number of platelets (×10^3^/μL) increased dramatically from pre- to post-transfusion (see Table 8).

The values of BUN, creatinine, ALT, and CRP were not correlated with the initial platelet counts and necessity of transfusion in the patients who received three platelet units (Table 9).

## 3. Discussion

The mean age of the patients with CKD and thrombocytopenia in this study was 63.96 years. There were statistical differences between the mean age values in men (59.75 years) and women (68.69 years) (t = −2.943, df = 98.191, *p* = 0.004 < α = 0.05). The results account for gender and age as risk factors for the progression of chronic kidney disease. Other associated diagnoses, which were seen as possible risk factors for thrombocytopenia in the studied patients with chronic kidney disease (CKD), included liver cirrhosis (CKD + LC) due to hepatitis C virus (22.12% of cases) and urosepsis: recurrent urinary tract infections and kidney abscesses (22.12% of cases) (CKD + S). We found that there was no association between the initial platelet counts, the necessity of transfusion, and associated diagnoses (Χ^2^_calc_ = 7.921, df = 4, *p* = 0.095 > α = 0.05).

Urinary stones or urinary obstructions were not present in our group of patients. Other causes, like SARS-CoV2 or other recent virus-related infections, drug-induced thrombocytopenia, or atypical hemolytic uremic syndrome (aHUS), were not diagnosed in our patients.

For the patients included in the study, the average of BUN was 174.51 mg/dL and the average of creatinine was 5.09 mg/dL. The median C reactive protein (CRP) level was 6.92 mg/L, with the CRP concentrations between 2 and 10 mg/L being considered markers of metabolic inflammation. The median patient platelet count at admission was 49.22 × 10^6^/μL. The values of BUN, serum creatinine, ALT, and CRP were not correlated with the initial platelet counts and the number of transfused platelet units. In other studies, elevated CRP levels were also positively associated with renal risk factors and diminished filtration or hyperfiltration, but not thrombocytopenia [9,10].

One important topic was whether platelets should be indicated for prophylactic versus therapeutic administration and the platelet threshold for transfusion. The prophylactic administration of platelets is reported to reduce the risk of bleeding and the odds of death in patients with hematologic malignancies, but few studies have explored this for other pathologies. In all cases included in our study, the platelet transfusions were prophylactic. The platelet thresholds for reversible bone marrow failure may be used as a general guide for other patient groups [11,12,13]. The threshold 20 × 10^3^ platelets/µL was considered for prophylactic transfusion and was preferred over a threshold of 10 × 10^3^ platelets/µL.

The number of platelets available for administration was smaller than the necessity, as platelet demand continuously increased in the last decade, more than blood donation. In 53.85% of cases, a low dose of platelets was administrated, with the initial platelet counts in these patients being over the threshold of 20 × 10^3^/µL.

In all cases, patients with platelet counts under the threshold of 20 × 10^3^/μL were administrated with at least two apheresis platelet units per patient (intermediary dose), and in 7.69%, a high dose of platelets was administrated, considering the additional risk factors for bleeding. 

The result of transfusions was satisfactory. With one single exception, the patients exceeded the critical threshold. After transfusions, most patients had over 50 × 10^3^ platelets/µL: there were 51 cases between 50 and 100 × 10^3^ platelets/µL and 39 cases over 100 × 10^3^ platelets/µL. Platelets increased with a mean no. of platelet score of 18.5 for one apheresis platelet unit ×10^3^/µL in CKD + LC, by 41.58 × 10^3^/µL in CKD + S, and by 33.3 × 10^3^/µL in CKD. Platelets increased with a mean no. of platelet score of 31.69 × 10^3^/µL in patients with CKD + LC, 37.86 × 10^3^/µL in patients with CKD + S, and 50.55 × 10^3^/µL in patients with CKD without associated diagnoses who received two apheresis platelet units. After the transfusion of three apheresis platelet units, the mean platelet count increased by 42.75 × 10^3^/µL in CKD + S and by 105.5 × 10^3^/µL in CKD.

Just in a single case (0.96%), the patient with platelet refractoriness had severe CKD associated with alloimmunization.

There were no post-transfusion adverse reactions declared. It was an important result of the measures applied to avoid any risky transfusion. We used leukoreduced blood products to avoid alloimmunization and we tested for irregular antibodies in all of the multiple blood-transfused patients.

Mortality during hospitalization in our study group was 8.65%, and 4.83% (5/104), due to severe bleedings. Severe, uncontrolled hemorrhages, in spite of platelet transfusions, are reported in this special category of patients, with multiple comorbidities [14,15]. Patients with chronic kidney disease develop severe thrombocytopenia as well as platelet dysfunction, which leads to bleeding. Most of them benefit from transfusions, but some cases are refractory to transfusion. Some patients develop alloimmunization to platelet antigens [16]. Epidemiological studies and case reports on hematologic disorders caused by renal disease help to develop treatment strategies [17,18,19]. Other studies have indicated the importance of the investigation of thrombocytopenia and platelet function to improve treatment strategies in patients at different stages of CKD (pre-dialysis or dialysis) [20,21,22]. This is to consider its consequences on kidney transplantation, such as the risk of developing de novo donor-specific antibodies (dnDSA), which is one of the main causes that limit long-term graft survival [23,24].

## 4. Material and Methods

This study was conducted on 104 patients with chronic kidney disease and thrombocytopenia who received platelet transfusion. The aim of this study was to assess the effectiveness of the administration of platelet concentrate and how patient prognosis was influenced by platelet transfusion therapy.

The patients were hospitalized in the nephrology ward of Constanta County Emergency Hospital Sf. Apostol Andrei. The period of the study was from January 2015 to December 2021. The complete blood count, blood urine nitrogen (BUN) and creatinine, ALT, and the C reactive protein were tested at admission.

The number of transfused platelet units and results in the platelet count was noted for each patient. The statistical analysis was performed using IBM SPSS statistics software, version 25. The data are presented as mean ± standard deviation (SD) for continuous variables in the case of symmetric distributions, median, and IQR (Interquartile range P_75_–P_25_) for continuous variables in the case of skewed distributions or percentages for categorical variables. The normality of the continuous data was estimated with Shapiro–Wilk tests of normality. For hypotheses testing, the following parametric and non-parametric tests were used: independent samples *t*-tests, one-way ANOVA tests, Wilcoxon signed-rank tests, independent samples Kruskal–Wallis H tests, independent samples median tests, and Chi-square tests for association. The significance level (α) was set at 0.05.

## 5. Conclusions

The patients with moderate-to-severe chronic kidney disease develop thrombocytopenia and a bleeding risk as a consequence. The indication of prophylactic transfusion was considered and the critical threshold was 20 × 10^3^ platelets/µL, being preferred over a threshold of 10 × 10^3^ platelets/µL. The initial platelet count and the necessity of platelet transfusion was not correlated with any associated diagnoses (liver cirrhosis and urosepsis), CRP, ALT, BUN, or creatinine levels.

Our study showed that the platelet transfusion clearly improved the prognosis in these patients, who exceeded the critical threshold with most of them overpassing 50 × 10^3^ platelets/µL. Severe hemorrhages were prevented in 95.17% of the cases, even though the number of platelets available for administration were under the necessities in many situations.

The collaboration between nephrology specialists and transfusion specialists is of a major importance in the management of platelet transfusion in CKD patients with severe thrombocytopenia. An important result of the measures applied to avoid any risky transfusion was that there were no post-transfusion adverse reactions with a maximum benefit for the patients.

The understanding behind platelet dysfunction in patients with moderate-to-severe chronic kidney disease causing either excessive clotting or bleeding remains elusive. More studies on the dynamic relationships between thrombosis, bleeding, and chronic inflammation associated with CKD progression are needed to improve the clinical management and outcomes for this vulnerable population that requires renal replacement therapy. Though red blood cell transfusions are well studied in pre-dialysis and dialysis CKD patients, studies regarding platelet transfusions in CKD patients are scarce and more research is needed to provide relevant evidence.

## Figures and Tables

**Figure 1 ijms-24-15895-f001:**
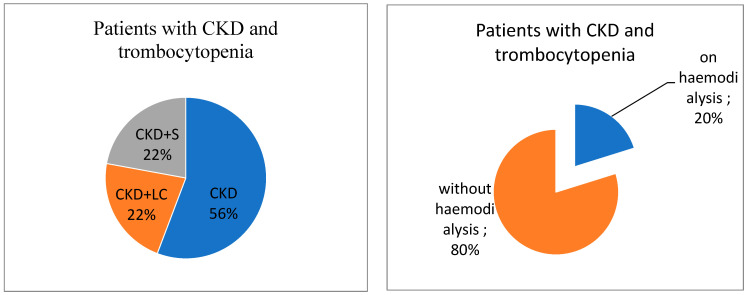
Patients with CKD-associated diagnoses and thrombocytopenia.

**Figure 2 ijms-24-15895-f002:**
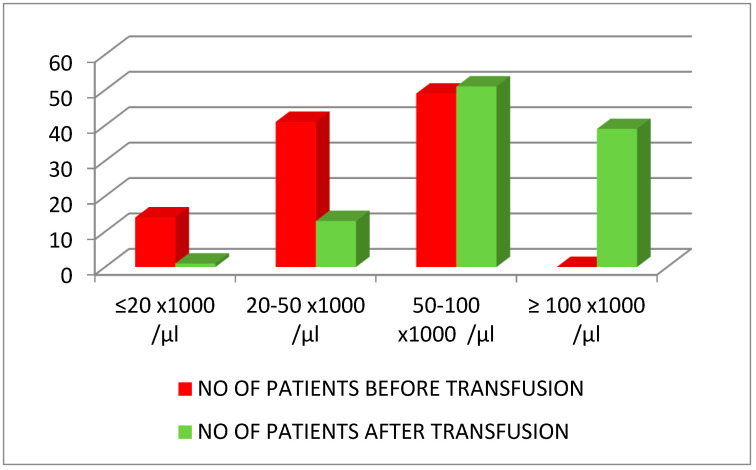
Results of platelet transfusion. Number of patients and platelet counts before and after transfusion.

**Figure 3 ijms-24-15895-f003:**
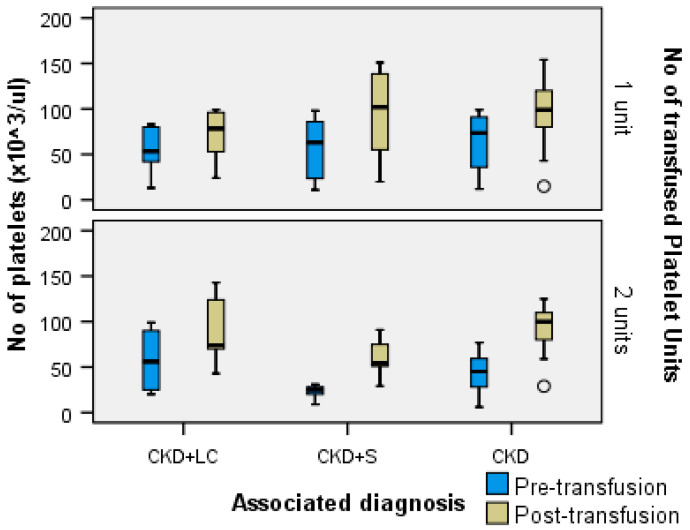
Box plot of no. of platelets (×10^3^/μL) for each associated diagnostic category for both pre-transfusion or post-transfusion time points according to the number of transfused platelet units.

**Figure 4 ijms-24-15895-f004:**
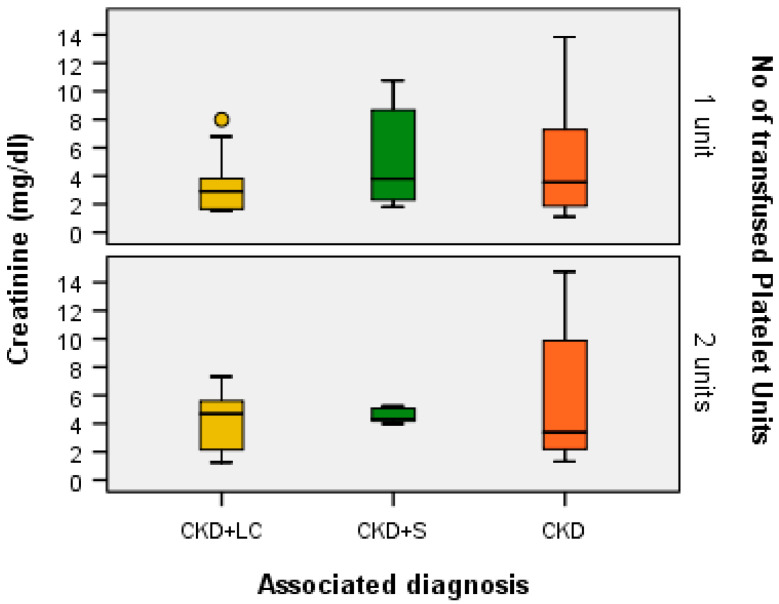
Box plot of creatinine (mg/dL) for each associated diagnostic category according to the number of transfused platelet units.

**Figure 5 ijms-24-15895-f005:**
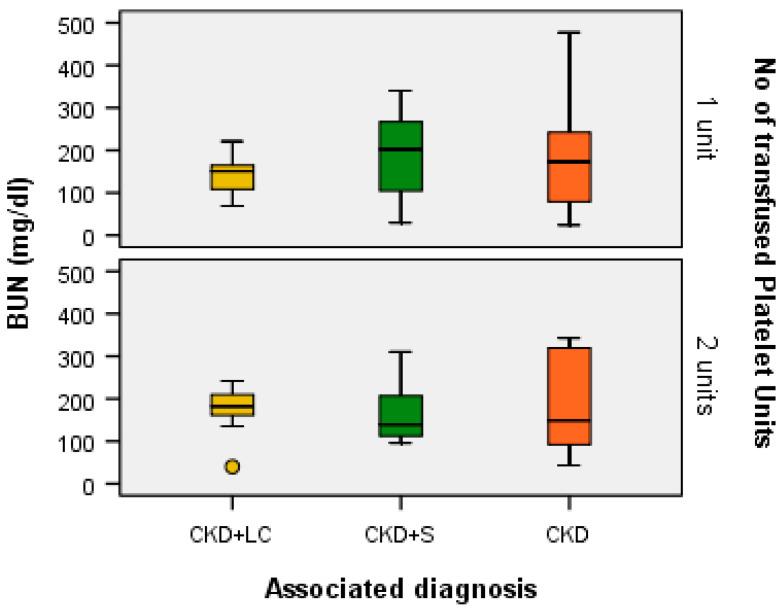
Box plot of BUN (mg/dL) for each associated diagnostic category according to the number of transfused platelet units.

**Figure 6 ijms-24-15895-f006:**
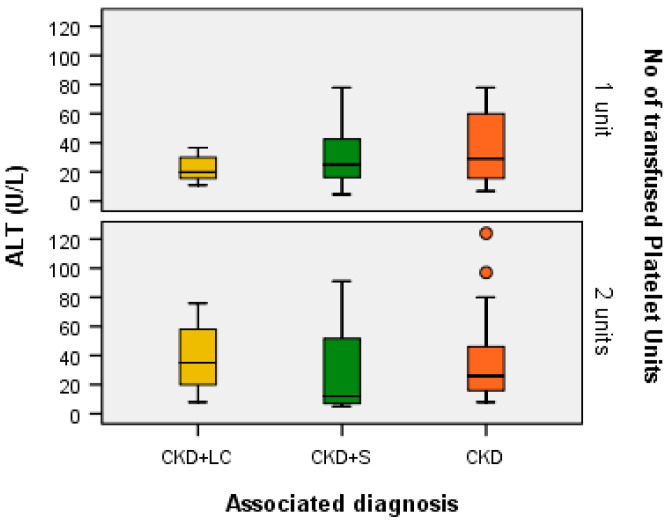
Box plot of ALT (U/L) for each associated diagnostic category according to the number of transfused platelet units.

**Figure 7 ijms-24-15895-f007:**
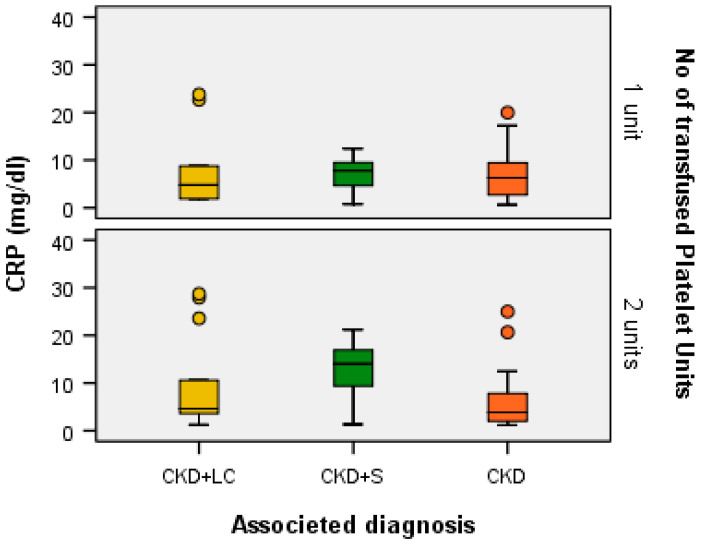
CRP (mg/dL) for each associated diagnostic category according to the number of transfused platelet units.

**Table 1 ijms-24-15895-t001:** Age and gender distribution in our study group.

Gender	No. of Patients	Mean Age (Years)	SD	Min	Max	P25	Median	P75
Men	55	59.75	17.86	22.00	90.00	48.00	64.00	71.00
Women	49	68.69	12.98	36.00	88.00	61.50	72.00	79.00

**Table 2 ijms-24-15895-t002:** Associated diagnosis. Gender Crosstabulation.

			Gender		Total
			Men	Women	
Associated diagnoses	CKD + LC	Count	11 (10.58%)	12 (11.54%)	23 (22.12%)
	CKD + S	Count	13 (12.50%)	10 (9.62%)	23 (22.12%)
	CKD	Count	31 (29.81%)	27 (25.96%)	58 (55.77%)
Total		Count	55 (52.88%)	49 (47.12%)	104 (100.00%)

**Table 3 ijms-24-15895-t003:** Associated diagnosis. Age Crosstabulation.

AD	N	Mean	SD	Min	Max	P25	Median	P75
CKD + LC	23	66.52	11.54	36.00	77.00	63.00	71.00	74.00
CKD + S	23	67.17	17.13	25.00	90.00	64.00	71.00	76.00
CKD	58	61.67	17.43	22.00	88.00	49.00	63.00	79.00

**Table 4 ijms-24-15895-t004:** No. of transfused Platelet Units.

		Frequency	Percentage
Valid	One Unit	56	53.85
	Two Units	40	38.46
	Three Units	8	7.69
	Total	104	100.00

**Table 5 ijms-24-15895-t005:** Results of platelet transfusion.

No. of PLT	≤20 × 10^3^/µL	20–50 × 10^3^/µL	50–100 × 10^3^/µL	≥100 × 10^3^/µL
No. of Patients before Transfusion	14	41	49	0
No. of Patients after Transfusion	1	13	51	39
Total n = 104				

**Table 6 ijms-24-15895-t006:** Associated diagnoses. No. of Transfused Platelet Units Crosstabulation.

			No. of Transfused Platelet Units			Total
			1 Unit	2 Units	3 Units	
Associated Diagnoses	CKD + LC	Count	10	13	0	23
		% of Total	9.62%	12.50%	0.00%	22.12%
	CKD + S	Count	12	7	4	23
		% of Total	11.54%	6.73%	3.85%	22.12%
	CKD	Count	34	20	4	58
		% of Total	32.69%	19.23%	3.85%	55.77%
Total		Count	56	40	8	104
		% of Total	53.85%	38.46%	7.69%	100.00%

**Table 7 ijms-24-15895-t007:** Platelet Count (×10^3^/μL) before and after transfusion.

No. TP	AD		N	Mean	SD	Min	Max	P25	Median	P75
One Unit	CKD + LC	Pre-transfusion	10	53.70	23.86	13.00	83.00	37.25	53.50	80.50
		Post-transfusion	10	72.20	26.97	24.00	99.00	49.25	78.50	96.50
	CKD + S	Pre-transfusion	12	56.25	34.60	11.00	98.00	17.75	63.00	86.50
		Post-transfusion	12	97.83	47.13	20.00	151.00	47.50	102.00	139.25
	CKD	Pre-transfusion	34	64.26	28.35	12.00	99.00	35.75	73.50	91.00
		Post-transfusion	34	97.56	31.93	15.00	154.00	79.00	99.00	121.00
Two Units	CKD + LC	Pre-transfusion	13	55.77	30.88	20.00	99.00	24.50	56.00	93.00
		Post-transfusion	13	87.46	34.23	43.00	143.00	61.50	74.00	126.50
	CKD + S	Pre-transfusion	7	23.00	7.46	9.00	31.00	20.00	25.00	29.00
		Post-transfusion	7	60.86	22.42	29.00	91.00	50.00	54.00	90.00
	CKD	Pre-transfusion	20	42.75	21.79	6.00	77.00	27.50	45.00	60.25
		Post-transfusion	20	93.30	24.73	29.00	125.00	79.00	100.00	111.50

**Table 8 ijms-24-15895-t008:** Statistics: No. of platelets (×10^3^/μL).

No. TP	AD		N	Mean	SD	Min	Max	P25	Median	P75
Three Units	CKD + S	Pre-transfusion	4	18.50	8.54	8.00	28.00	10.00	19.00	26.50
		Post-transfusion	4	61.25	36.70	29.00	95.00	29.25	60.50	94.00
	CKD	Pre-transfusion	4	25.00	9.42	16.00	35.00	16.50	24.50	34.00
		Post-transfusion	4	130.50	22.01	110.00	151.00	110.75	130.50	150.25

**Table 9 ijms-24-15895-t009:** Statistics: Creatinine (mg/dL), BUN (mg/dL), ALT (U/L), and CRP (mg/dL).

No. TP	AD		N	Mean	SD	Min	Max	P25	Median	P75
Three Units	CKD + S	Creatinine (mg/dL)	4	4.56	2.74	1.76	6.94	1.98	4.77	6.93
		BUN (mg/dL)	4	292.70	181.41	132.00	472.39	134.25	283.20	460.64
		ALT (U/L)	4	28.94	17.85	13.00	45.23	13.25	28.77	44.81
		CRP (mg/dL)	4	5.68	4.08	2.10	9.45	2.13	5.58	9.33
	CKD	Creatinine (mg/dL)	4	2.16	1.22	1.40	3.98	1.42	1.64	3.43
		BUN (mg/dL)	4	100.33	79.84	56.31	220.00	57.73	62.50	180.75
		ALT (U/L)	4	85.37	47.11	25.47	124.00	36.60	96.00	123.50
		CRP (mg/dL)	4	4.67	5.05	1.40	12.20	1.67	2.54	9.79

## Data Availability

Not applicable.

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
