# Peer review of "Efficiency of Platelet Transfusion in Patients with Moderate-to-Severe Chronic Kidney Disease and Thrombocytopenia"

_ijms, 2023, doi:10.3390/ijms242115895_

Round 1
Reviewer 1 Report
Comments and Suggestions for Authors
This is a descriptive study of CKD patients receiving transfusion with platelets.
Although the topic is of some interest, the writing and presentation of results are flawed.
1) In the abstract: "... that for 1 transfused platelet unit 24 the distribution of number of platelets (x103/μl) is the same across categories of associated diagnosis." What is associated diagnosis is not defined and describe until the results section. This type of gaps in the writing makes it difficult to understand.
2) The authors should consider stating clearly the goal of the study /hypothesis in the introduction.
3) The title does not accurately reflect the findings.
4) It would be clear to have a section in the results section given the descriptive of the study patients; e.g. one table summarizing all factors.
This should then be followed by a subsection results of platelet transfusion (Section 3.4) and following the outline of the study specifics/hypotheses typically presented in the introduction.
Comments on the Quality of English Language
The paper will benefit from an professional English editor.
Frequency of errors, awkward phrases etc. too long to list.
Author Response
Thank you for the review and valuable suggestions!
- Indeed, a clarification was necessary. We added in the abstract the categories of associated diagnoses:” associated diagnoses, seen as possible risk factors for thrombocytopenia: liver cirrhosis and urosepsis.”
- We reformulated and clearly stated the aim of the study in the introduction: This is a descriptive study of thrombocytopenic CKD patients requiring platelet transfusions. The aim of the study was to establish if the administration of platelet concentrate was efficient and how the patients’ prognosis was influenced by platelets transfusion therapy. An important issue was to determine the platelet count threshold for prophylactic administration in patients with moderate-to-severe chronic kidney disease and thrombocytopenia, related to their higher risk of bleeding.
- We made a change in the title, to better reflect the results of or study: Efficiency of Platelet Transfusion in Patients with Moderate-to-Severe Chronic Kidney Disease and Thrombocytopenia
- We tried to improve the presentation of the results, but only by correcting the existing tables and we changed 3.3. and 3.4. 3. Transfusion requirements. 3.4. Results of platelet transfusion.
- We made English language correction - English-Editing-Certificate72357
Reviewer 2 Report
Comments and Suggestions for Authors
In the manuscript entitled "Platelet abnormalities and transfusion requirement in patients with moderate-to-severe chronic kidney disease" Sevigean Ali et al. provide evidence regarding platelet abnormalities in the setting of CKD. The study focuses on an interesting theme and could give a glimpse at the association between transfused platelet units and patients’ condition post-transfusion.
However, this Reviewer has some comments that must be addressed, as follow:
1. Materials and Methods section: It would be more fitted if the aim of the study was placed at the end of the introduction.
2. The first Tables that refer mainly to the age and sex of the participants can perhaps be merged or their information transferred to the text to facilitate the reader.
3. Please mention all tables and figures in the text before their appearance throughout the manuscript
4. Could the authors please add marks of significance (where applicable) in the figures?
5. Results section: Lines 194-197 ““Low platelet counts have been identified as a surrogate marker for poor prognosis in septic patients”[10]. In all cases the platelet transfusions were prophylactic. Adverse reactions: There were no adverse reactions.” seem to be out of context. Please review this part.
6. Discussion section: This reviewer’s major concern is that the discussion section mainly reads like a repetition of the results. The authors do not discuss their results nor they support the novelty of their findings. The specific section needs extensive re-structure.
7. In general, an extensive proof-reading for grammatical mistakes throughout the manuscript is needed.
Comments on the Quality of English LanguageIn general, an extensive proof-reading for grammatical mistakes throughout the manuscript is needed.
Author Response
Response to the 2nd reviewer
Thank you for the review and valuable suggestions!
- We have added the aim of the study at the end of introduction, clearly stated: This is a descriptive study of thrombocytopenic CKD patients requiring platelet transfusions. The aim of the study was to establish if the administration of platelet concentrate was efficient and how the patients’ prognosis was influenced by platelets transfusion therapy. An important issue was to determine the platelet count threshold for prophylactic administration in patients with moderate-to-severe chronic kidney disease and thrombocytopenia, related to their higher risk of bleeding.
- We have improved Table 1: Distribution of age and gender, to facilitate the reader
- We’ve made the corrections and mentioned all tables and figures in the text before their appearance in the manuscript.
- We tried to improve the explanation for the figures.
- Indeed it was out of context. We reviewed it . The comments about adverse reactions are now in Discussion section : There were no post-transfusion adverse reactions declared. It was an important result of the measures applied to avoid any risky transfusion. We used leukoreduced blood products to avoid alloimmunisation and we tested for irregular antibodies all multiple blood transfused patients.
- We have reviewed and extensively restructured the Discussion section.
- We have made English language correction - English-Editing-Certificate72357
Round 2
Reviewer 1 Report
Comments and Suggestions for Authors
Made some acceptable revisions.
Comments on the Quality of English LanguageNA.
Reviewer 2 Report
Comments and Suggestions for Authors
The authors have addressed my comments. I have nothing more to add.